# Current Treatment Strategies and Nanoparticle-Mediated Drug Delivery Systems for Pulmonary Arterial Hypertension

**DOI:** 10.3390/ijms20235885

**Published:** 2019-11-23

**Authors:** Kazufumi Nakamura, Satoshi Akagi, Kentaro Ejiri, Masashi Yoshida, Toru Miyoshi, Norihisa Toh, Koji Nakagawa, Yoichi Takaya, Hiromi Matsubara, Hiroshi Ito

**Affiliations:** 1Department of Cardiovascular Medicine, Okayama University Graduate School of Medicine, Dentistry and Pharmaceutical Sciences, Okayama 700-8558, Japan; akagi-s@cc.okayama-u.ac.jp (S.A.); eziken82@gmail.com (K.E.); masashiyoshid@gmail.com (M.Y.); miyoshit@cc.okayama-u.ac.jp (T.M.); Norihisa.Toh@okayama-u.ac.jp (N.T.); takayayoichi@yahoo.co.jp (Y.T.); itomd@md.okayama-u.ac.jp (H.I.); 2Division of Cardiology, National Hospital Organization Okayama Medical Center, Okayama 701-1192, Japan; matsubara.hiromi@gmail.com

**Keywords:** pulmonary arterial hypertension, prostaglandin I_2_, nitric oxide, endothelin

## Abstract

There are three critical pathways for the pathogenesis and progression of pulmonary arterial hypertension (PAH): the prostacyclin (prostaglandin I_2_) (PGI_2_), nitric oxide (NO), and endothelin pathways. The current approved drugs targeting these three pathways, including prostacyclin (PGI_2_), phosphodiesterase type-5 (PDE5) inhibitors, and endothelin receptor antagonists (ERAs), have been shown to be effective, however, PAH remains a severe clinical condition and the long-term survival of patients with PAH is still suboptimal. The full therapeutic abilities of available drugs are reduced by medication, patient non-compliance, and side effects. Nanoparticles are expected to address these problems by providing a novel drug delivery approach for the treatment of PAH. Drug-loaded nanoparticles for local delivery can optimize the efficacy and minimize the adverse effects of drugs. Prostacyclin (PGI_2_) analogue, PDE5 inhibitors, ERA, pitavastatin, imatinib, rapamycin, fasudil, and oligonucleotides-loaded nanoparticles have been reported to be effective in animal PAH models and in vitro studies. However, the efficacy and safety of nanoparticle mediated-drug delivery systems for PAH treatment in humans are unknown and further clinical studies are required to clarify these points.

## 1. Introduction

Pulmonary arterial hypertension (PAH) is a progressive disease caused by vasoconstriction and remodeling of the pulmonary vasculature [1,2,3]. Recent development of PAH-targeted drugs has resulted in improvement of prognosis and quality of life in patients with PAH [4,5]. However, long-term survival of patients with PAH is still suboptimal. Therefore, new treatment is thought to be needed.

## 2. Medical Treatment of Pulmonary Arterial Hypertension (PAH)

Modern development of drugs for PAH focus on three pathways, namely the prostacyclin (prostaglandin I_2_) (PGI_2_), nitric oxide (NO), and endothelin pathways [4,6]. Impaired production of vasodilators such as PGI_2_ and NO, along with over-expression of vasoconstrictors such as endothelin-1 are critical for the pathogenesis and progression of PAH. Drugs targeting the three pathways, including prostacyclin (PGI_2_), endothelin receptor antagonists (ERAs), phosphodiesterase type-5 (PDE5) inhibitors, and a soluble guanylate cyclase (sGC) stimulator, are currently available and have been shown to be effective (Figure 1) [4,7,8,9,10,11].

### 2.1. Prostacyclin (PGI_2_)

Prostacyclin (PGI_2_) is released by endothelial cells and activates adenylate cyclase via the prostaglandin I_2_ receptor (IP) in pulmonary artery smooth muscle cells (PASMCs). Activated adenylate cyclase catalyzes the conversion of adenosine triphosphate (ATP) to 3’5’-cyclic AMP (cAMP), which activates protein kinase A (PKA). PKA goes on to promote the phosphorylation of myosin light chain kinase, which leads to smooth muscle relaxation and vasodilation.

Prostacyclin, also known as epoprostenol; synthetic prostacyclin analogues including treprostinil, iloprost, and beraprost, and a selective prostacyclin receptor (IP receptor) agonist, selexipag, are used for the treatment of PAH. The efficacy of continuous intravenous epoprostenol therapy has been tested in three unblinded randomized clinical trials (RCTs) in patients with idiopathic PAH (IPAH) [12,13] and in patients with pulmonary hypertension (PH) owing to the scleroderma spectrum of disease, WHO-functional class (WHO-FC) III or IV despite optical medical therapy [14]. In RCTs, epoprostenol treatment improves symptoms, exercise capacity, and hemodynamics, and reduces mortality of patients with IPAH [9,13]. Selexipag, an oral IP receptor agonist, significantly reduced a composite of death from any cause or a complication related to PAH (GRIPHON trial) [15].

### 2.2. Phosphodiesterase Type 5 (PDE5) Inhibitors and Soluble Guanylate Cyclase (sGC) Stimulator

Nitric oxide (NO) released from vascular endothelium activates the enzyme guanylate cyclase, which results in increased levels of cyclic guanosine monophosphate (cGMP), leading to smooth muscle relaxation in pulmonary arteries. The critical role of the NO-sGC-cGMP pathway in regulating pulmonary vascular tone is demonstrated by the dysregulation of NO production, sGC activity, and cGMP degradation in PH.

PDE5 inhibitors inhibit the degradation of cGMP by PDE5. Sildenafil and tadalafil have been used for the treatment of PAH. Riociguat is a pharmacological agent that directly stimulates sGC, both independently of NO and in synergy with NO. Riociguat treats two forms of PH: chronic thromboembolic pulmonary hypertension (CTEPH) and PAH.

### 2.3. Endothelin Receptor Antagonists (ERAs)

Endothelin type A receptor (ET_A_) and type B receptor (ET_B_) are G protein-coupled receptors whose activation results in elevation of intracellular-free calcium. Endothelin-1 (ET-1) abluminally released from vascular endothelium causes the underlying smooth muscle to contract, mainly via ET_A_. ERAs are drugs that block endothelin receptors. Selective ET_A_ receptor antagonists, ambrisentan, and dual antagonists that affect ET_A_ and ET_B_, including bosentan and macitentan, are used for the treatment of PAH. In an RCT (SERAPHIN trial) adopting clinical aggravation as the composite primary endpoint, which consists of a first event related to PAH or death from any cause, the macitentan 10 mg treatment group showed significant improvement compared with the placebo group [16].

### 2.4. Combination Therapy

Recently, most experts in PH have empirically used a combination of PAH-targeted drugs for treating patients [5,17,18]. The Japan PH registry showed that initial upfront combination therapy was associated with improvement in hemodynamic status [5]. The AMBITION study showed that using the upfront combination therapy with ambrisentan and tadalafil reduced the risk of the primary endpoint of the first event of clinical failure (a composite of death, hospitalization for worsening pulmonary arterial hypertension, disease progression, or unsatisfactory long-term clinical response) by 50% compared with each individual treatment [19].

### 2.5. Current Status and Future Perspectives

As noted above, current approved drugs that target the three described pathways have been shown to be effective. However, PAH remains a severe clinical condition despite the publication of 41 RCTs. Many researchers are working on finding novel therapeutic targets and potential drugs for PAH [20,21].

Furthermore, the full therapeutic potential of the drugs targeting the three pathways is reduced by medication, patient non-compliance, and side effects, meaning that the long-term survival of patients with PAH remains suboptimal. To address these problems, several novel therapeutic strategies for treating PAH, such as nanoparticle-mediated drug delivery systems (nano-DDS), have been proposed.

## 3. Nanoparticle-Mediated Drug Delivery Systems (Nano-DDS)

Nanoparticles (NPs) have been used in numerous novel delivery systems for the transport of drugs to target organs [22,23,24,25]. NPs are taken up by the target organ because of their small size, which allows them to permeate into tissue and be retained. Drug release from NPs can be controlled by the NP composition. Thus, drug-loaded NPs for local delivery can optimize the efficacy and minimize the side effects of drugs. All drugs have potential toxicity that may limit their safe dose and thereby their therapeutic efficacy. The use of a nano-DDS can enhance the efficacy and safety of therapeutic agents, and overcome drawbacks such as toxicity, low water solubility, and poor bioavailability [23].

Nano-DDS can be composed of a variety of materials and structures, including micelles, liposomes, dendrimers, and polymers [23,25]. Polymeric nanospheres are formed by the assembly of macromolecular polymers and can contain hydrophilic and hydrophobic therapeutic agents. Two polymers, polylactide (PLA) and poly (lactide-co-glycolic acid) (PLGA), are used for the synthesis of FDA-approved polymeric biodegradable nano-DDS (Figure 2) [23].

## 4. Nano-DDS for PAH Treatment

Treatment of PAH with vasodilators such as prostacyclin (PGI_2_), ERAs, PDE5 inhibitors, and an sGC stimulator has been effective [4,7,8,9,10,11], however PAH is still a fatal disorder in some patients. Intravenous administration of prostacyclin or systemic administration of imatinib causes several adverse events and complications, therefore several novel therapeutic strategies for PAH, including nano-DDS, have been proposed [24,25,26]. Nano-DDS for lung treatment could optimize the efficacy and minimize the side effects of drugs.

Prostacyclin (PGI_2_) analogue [27,28,29,30], PDE5 inhibitors [31,32], ERA [33], oligonucleotides [34,35], pitavastatin [36,37], imatinib [38], rapamycin [39], and fasudil [40]-loaded NPs have been reported to be effective in animal PAH models and in vitro studies (Table 1). All of the data in the studies were obtained in animal models of PAH or in vitro studies and have not yet been translated to human trials. The efficacy and safety of nano-DDS for PAH in humans are unknown and further studies are needed to clarify these issues.

### 4.1. Prostacyclin (PGI_2_) Analogue-Loaded Nanoparticles (NPs)

The production of endogenous prostacyclin (PGI_2_) is suppressed in pulmonary arteries of patients with PAH. PGI_2_ synthase expression has been reported to be reduced in lung tissues from patients with severe pulmonary hypertension (PH) [41]. Prostacyclin replacement therapy by infusion of epoprostenol sodium, a prostacyclin (PGI_2_), is one of the best available treatments for severe PAH. In RCTs, continuous intravenous epoprostenol therapy improves symptoms, exercise capacity, and hemodynamics, and reduces mortality in patients with IPAH. We reported that high-dose epoprostenol therapy (>40 ng/kg/min) resulted in marked hemodynamic improvement in patients with PAH [7,8,42]. Compared with the baseline state, high-dose epoprostenol therapy reduced mean pulmonary arterial pressure (mPAP) by 30% and pulmonary vascular resistance (PVR) by 68%, and increased cardiac index by 89% and SvO_2_ by 19% [7].

However, there are several adverse events and complications such as headache, hypotension, and catheter-related infection in intravenous prostacyclin replacement therapy. Systemic administration of prostacyclin can induce headache, flushing, and in some cases, severe hypotension at the start of therapy. One of the most serious complications is catheter-related infections. These limitations could potentially be resolved by the development of an alternative system to target prostacyclin delivery to the pulmonary vasculature without using a central venous catheter.

Beraprost-loaded NPs: Nano-DDS using prostacyclin-loaded NPs for lung delivery might optimize the efficacy and minimize the side effects of drugs. Therefore, we developed PLGA-NPs loaded with beraprost, a prostacyclin analogue (beraprost-NPs) and investigated the efficacy and safety of intratracheal administration of beraprost-NPs in Sugen-hypoxia-normoxia and monocrotaline (MCT) rat models of PAH [28]. Single intratracheal administration of beraprost-NPs significantly reduced right ventricular pressure, right ventricular hypertrophy, and pulmonary artery muscularization in both rat models. Furthermore, beraprost-NPs significantly improved the survival rate in the MCT rat model.

Intravenous administration (once per week) of poly(lactide) (PLA)-NPs loaded with beraprost also prevent pulmonary arterial remodeling and right ventricular hypertrophy in MCT-induced rat PAH model and hypoxia-induced mouse PAH model [27].

Treprostinil-loaded NPs: Trepostinil was also chemically modified to be an alkyl prodrug (TPD) which was then packaged into a lipid nanoparticle (LNP) carrier. The TPD-LNP demonstrated approximately lower treprostinil plasma concentration compared to inhaled treprostinil solution while maintaining an extended vasodilatory effect in a rat model of hypoxia-induced pulmonary vasoconstriction [29].

Iloprost-loaded NPs: Jain et al. synthesized liposomal nanoparticles tailored for the prostacyclin analogue iloprost. Iloprost-NPs exhibited significantly enhanced vasodilation of isolated tertiary pulmonary arteries in mice [30].

### 4.2. PDE5 Inhibitor-Loaded NPs

Sildenafil-loaded NPs: Sildenafil, a PDE5 inhibitor, exhibits the preventative effects of pulmonary artery remodeling, however, conventional sildenafil has shown limited treatment efficacy owing to its poor accumulation in pulmonary arteries. Li et al. reported that glucuronic acid (GlcA)-modified liposomes with encapsulated sildenafil improved the delivery of sildenafil to aberrant over-proliferative PASMCs by targeting glucose transport-1 (GLUT-1) and inhibited the remodeling of pulmonary arteries in MCT-induced PH model rats [31].

Tadalafil-loaded NPs: Dry powder inhaler containing tadalafil-loaded PLGA NPs are developed [32].

### 4.3. ERA-Loaded NPs

Bosetntan-loaded NPs: Bosentan monohydrate-loaded ε-polycaprolactone (PLC) NPs are made by electrospraying technique [33].

### 4.4. Others

Pitavastatin-loaded NPs: 3-Hydroxy-3-methylglutanyl coenzyme A (HMG-CoA) reductase inhibitors (statins) are well-tolerated drugs, which are widely used for lowering serum low-density lipoprotein cholesterol levels. Statins have pleiotropic effects, including improvement of endothelial cell functions, inhibition of vascular smooth muscle cell growth, and anti-inflammatory effects [43,44]. Simvastatin, an HMG-CoA reductase inhibitor, significantly inhibited PDGF-induced cell proliferation and migration of PASMCs from patients with idiopathic PAH [45]. In addition, simvastatin has been shown to attenuate pulmonary vascular remodeling by increasing vascular SMCs, and improve prognosis in rats with MCT-induced PH [46,47].

Pitavastatin, a statin, also has inhibitory effects on proliferation of PASMCs in vitro. Intratracheal administration of PLGA-NPs loaded with pitavastatin (pitavastatin-NPs) attenuated the development of MCT-induced rat PAH model [36]. Furthermore, a single intratracheal treatment with pitavastatin-NP three weeks after MCT injection induced regression of PAH and improved the survival rate [36].

A phase I investigator-initiated clinical trial to test the safety, tolerability, and pharmacokinetics of PLGA nanoparticle-mediated delivery of pitavastatin (NK-104-NP) has been completed (UMIN000014940 and UMIN000019189) [48]. The results showed that the pitavastatin-loaded NPs (NK-104-NP) exhibited dose-dependent pharmacokinetics and were well tolerated with no significant AEs in healthy volunteers. Now phase II clinical trial is conducted (jRCT2031180075).

## 5. Future Challenges and Possible Nanomedicine-Based DDS Solutions

The future challenge is to synthesize and develop novel nanomedicine-based DDS for the treatment of PAH that specifically target the pulmonary vascular cells, without exhibiting any toxicity in other cells.

### 5.1. Targeting

In nanomedicine-based DDS for the treatment of cancer, NPs can target leaky tumor vasculature through the enhanced permeability and retention (EPR) effect; however, the non-specificity towards healthy cells and tissues is still high. To overcome this non-specific toxicity of NPs toward healthy cells, several targeting agents have been used to enhance the tumor specific uptake and activity of NPs (active targeting) [49]. Because αvβ3 integrin, a cell surface receptor, is dramatically upregulated during neovascularization in the case of most cancers, targeting of integrin receptors by either αvβ3 integrin antibody or anti-angiogenic peptides (RGD/NGR), showed some exciting therapeutic results.

In PAH, there is an inhomogeneous distribution of vascular remodeling or cell viability conditions in small pulmonary arteries. Therefore, patients with intractable PAH require cell-specific or vascular compartment-specific therapies. Li et al. reported efficacious targeted delivery of sildenafil to PASMCs with glucuronic acid (GlcA)-modified liposomes in an MCT-induced PH model in rats [31]. They first demonstrated the overexpression of GLUT-1, a glucose transporter, on PASMCs in an MCT-induced PH rat model. The overexpressed GLUT-1 provided a potential target for GlcA-modified liposomes.

### 5.2. Risk Assessment Strategy for Novel Nanomaterials

Establishing a risk assessment strategy for novel nanomaterials is important to reduce the number of animal experiments. Conventional risk assessment techniques for chemical agents such as the category approach, read-across, and quantitative structure–activity relationship (QSAR) models may be useful [50,51].

### 5.3. Novel Therapeutic Targets and Potential Drugs for PAH

In addition to traditional therapy, emerging pre-clinical data and clinical trials suggest novel therapeutic targets and potential drugs for PAH (Table 2) [20,21]. Among those potential drugs, imatinib and fasudil-loaded NPs have been reported [38,40]. NPs are expected to provide novel delivery systems for the administration of potential drugs.

## 6. Summary

In this review, we summarized current treatment strategies by drugs targeting the three pathways and nano-DDS for PAH. Drug-loaded NPs for local delivery may optimize the efficacy and minimize the side effects of drugs targeting the three pathways and other potential drugs. However, the efficacy and safety of nano-drug delivery systems for PAH in humans are unknown. Further clinical studies are needed to clarify these points.

## Figures and Tables

**Figure 1 ijms-20-05885-f001:**
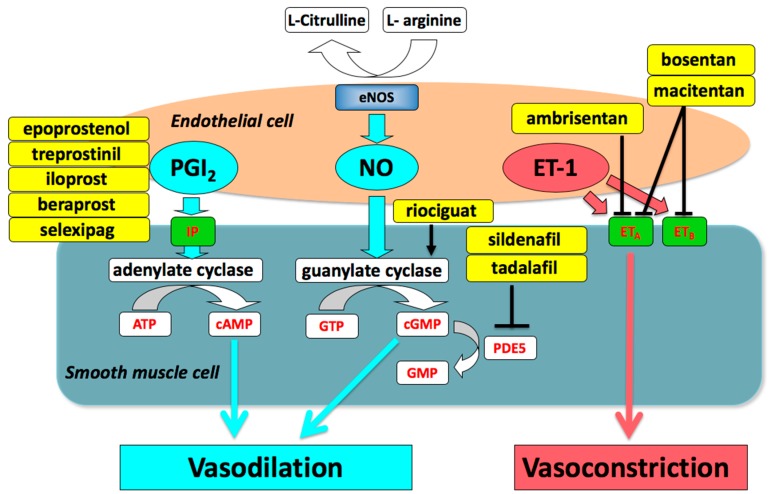
Drugs targeting the three pathways involved in the pathogenesis of pulmonary arterial hypertension. eNOS, endothelial nitric oxide synthase; PGI_2_, prostaglandin I_2_; NO, nitric oxide; ET-1, endothelin-1; IP, prostaglandin I_2_ receptor; ET_A_, endothelin type A receptor; ET_B_, endothelin type B receptor; ATP, adenosine triphosphate; cAMP, cyclic adenosine monophosphate; GTP guanosine triphosphate; cGMP, cyclic guanosine monophosphate; PDE5, phosphodiesterase type 5.

**Figure 2 ijms-20-05885-f002:**
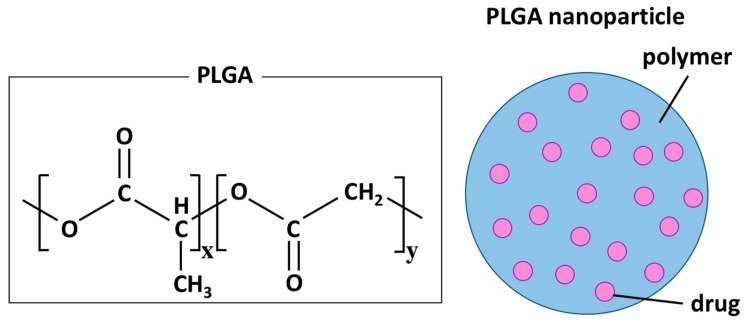
Chemical structure of poly (lactide-co-glycolic acid) (PLGA) and a schematic representation of a polymer nanoparticle. x and y represent the number of each unit in PLGA.

**Table 1 ijms-20-05885-t001:** Nanoparticle-mediated drug delivery systems for PAH treatment.

Drug	Delivery System	Animal Model	Route of Admin.	Refs
Prostacyclin analogues				
Beraprost	Polymer (PLA and PEG-PLA)	MCT-induced rat model	Intravenous	[27]
Beraprost	Polymer (PLA and PEG-PLA)	Hypoxia mouse model	Intravenous	[27]
Beraprost	Polymer (PLGA)	MCT-induced rat model	Intratracheal	[28]
Beraprost	Polymer (PLGA)	Sugen/hypoxia rat model	Intratracheal	[28]
Treprostinil	Lipid nanoparticle	Hypoxia rat model	Inhalation	[29]
Iloprost	Liposome	Isolated PA of mice		[30]
PDE5 inhibitors				
Sildenafil	GlcA-modified liposome	MCT-induced rat model	Intravenous	[31]
Tadalafil	Polymer (PLGA)	In vitro study		[32]
ERA				
Bosentan	Polymer (PCL)	In vitro study		[33]
Others				
Pitavastatin	Polymer (PLGA)	MCT-induced rat model	Intratracheal	[36]
Pitavastatin	Polymer (PLGA)	MCT-induced rat model	Intravenous	[37]
Imatinib	Polymer (PLGA)	MCT-induced rat model	Intratracheal	[38]
Rapamycin	Polymer (PEG-PCL)	MCT-induced rat model	Intravenous	[39]
Fasudil	Liposome	MCT-induced rat model	Inhalation	[40]
Oligonucleotides				
NF-kB decoy	Polymer (PEG-PLGA)	MCT-induced rat model	Intratracheal	[34]
AntimiRNA-145	Liposome	Sugen/hypoxia rat model	Intravenous	[35]

ERA, endothelin receptor antagonist; GlcA, glucuronic acid; MCT, monocrotaline; NF-kB, nuclear factor kappaB; PA, pulmonary artery; PAH, pulmonary arterial hypertension; PCL, poly(ε-caprolactone); PDE5, phosphodiesterase type-5; PEG-PCL, poly-(ethyleneglycol)-block-poly(ε-caprolactone); PEG-PLA, poly-(ethyleneglycol)-block-PLA; PEG-PLGA, poly-(ethyleneglycol)-block-PLGA; PLA, poly(lactide) homopolymer; PLGA, polylactide-glycolide.

**Table 2 ijms-20-05885-t002:** Novel pathways, therapeutic targets, and potential drugs for PAH.

Pathways	Therapeutic Targets	Potential Drugs
Growth factor	PDGF, EGF, FGF and VEGF	tyrosine kinase inhibitors
		Imatinib [52]
Inflammation	IL-6	tocilizumab [53]
	RAGE	RAGE aptamer, AS-1 [54]
	Nrf2 and NFkB	bardoxolone methyl [21]
BMPR-II	BMPR2 and sma-9	tacrolimus [55]
		ataluren [56]
Metabolic modulators	glucose oxidation	dichloroacetate [57]
Neurohormonal activation	sympathetic nerve system	β-blockers [58]
DNA damage	BRCA1 and PARP	olaparib [21]
Epigenetic modification	HDAC6	tubastatin A [59]
Vasoactive mediators	5HT	5HT-receptor antagonists [60]
	rho A/ROCK	fasudil [61]
	adrenomedullin	adrenomedullin [62]
	Apelin	apelin [63]

PDGF, platelet-derived growth factor; EGF, epidermal growth factor; FGF, fibroblast growth factor; VEGF, vascular endothelial growth factor; IL, interleukin; RAGE, receptor for advanced glycation end products; BMPR, bone morphogenetic protein receptor; BRCA1, breast cancer susceptibility gene I; PARP, Poly(ADP-ribose) polymerase 1; HT, hydroxytryptamine; ROCK, rho-kinase.

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
