# Peer review of "Current Treatment Strategies and Nanoparticle-Mediated Drug Delivery Systems for Pulmonary Arterial Hypertension"

_ijms, 2019, doi:10.3390/ijms20235885_

Round 1

Reviewer 1 Report

Dear author,

I have read the article entitled, 'Nanoparticle-mediated Drug Delivery System for Pulmonary Arterial Hypertension Treatment' by Nakamura et al, with high interest. The article describes nicely about the recent development of nanomedicine in the drug delivery to pulmonary arterial hypertension. However, the article needs some revisions before acceptance.

1. The author didn't discuss about the current state of Pulmonary arterial hypertension (PAH),  their conventional treatment, challenges and alternative treatment options. The authors are suggested to make a point before starting nanoDDS.

2. The introduction and summary portion is very small and provide very little insight of the current state of the disease and solutions. Author should elaborate these portions with more current references.

3. Author should include a section about the challenges and possible solutions of nanomedicine based DDS for the treatment of Pulmonary arterial hypertension (PAH) before the conclusion section. Please refer the following paper as a reference, (Nanoscale, 2016, 8, 12444-12470)

4. What are the clinical trials ongoing on the nanomedicine based treatment of Pulmonary arterial hypertension (PAH). Authors should make a table for this.

5. Overall, the article needs minor English and grammatical corrections throughout the text.

Author Response

Dear Professor Mihai V. Putz,

We would like to submit our manuscript entitled, “Nanoparticle-mediated Drug Delivery System for Pulmonary Arterial Hypertension Treatment” to special issue entitled "Nano-Materials and Methods", International Journal of Molecular Sciences.

The editor and the reviewer kindly pointed out several parts in our previous manuscript that should be clarified, and we have revised these parts.

The answers to the reviewer’s comments are as follows.

Reviewer: 1

We greatly appreciate the reviewer’s comments.

Comment: 1.The author didn't discuss about the current state of Pulmonary arterial hypertension (PAH), their conventional treatment, challenges and alternative treatment options. The authors are suggested to make a point before starting nanoDDS.

Answer: In accordance with the reviewer’s comment, we added the new paragraph: "2. Treatment of PAH"and Figure 1 (page 1-4 in the new manuscript).

Comment: 2. The introduction and summary portion is very small and provide very little insight of the current state of the disease and solutions. Author should elaborate these portions with more current references.

Answer: We appreciate the reviewer’s comment. In accordance with the reviewer’s comment,we added the two paragraphs: "2. Treatment of PAH" (page 1-4 in the new manuscript) and "5. Future challenges and possible nanomedicine-based DDS solutions"(page 8 in the new manuscript) and 17references.

Comment: 3. Author should include a section about the challenges and possible solutions of nanomedicine based DDS for the treatment of Pulmonary arterial hypertension (PAH) before the conclusion section. Please refer the following paper as a reference, (Nanoscale, 2016, 8, 12444-12470).

Answer: In accordance with the reviewer’s comment, we added the paragraphs: "5. Future challenges and possible nanomedicine-based DDS solutions"andreference #64(page 8 in the new manuscript).

Comment: 4. What are the clinical trials ongoing on the nanomedicine based treatment of Pulmonary arterial hypertension (PAH). Authors should make a table for this.

Answer: We appreciate the reviewer’s comment. Now one trial of pitavastatin -loaded NPs is conducted as far as we know. In accordance with the reviewer’s comment,we added the following sentences: "Now phase II clinical trial is conducted(jRCT2031180075)."(page 7,line 25in the new manuscript).

Comment: 5. Overall, the article needs minor English and grammatical corrections throughout the text.

Answer: We appreciate the reviewer’s comment. In accordance with the reviewer’s comment, we have our manuscript corrected. We added the following sentences: "We thank Sarah Dodds, PhD, from Edanz Group (www.edanzediting.com/ac) for editing a draft of this manuscript."(page 9,in Acknowledgmentsof the new manuscript).

Editor: 1

Comment: A somehow "thin" manuscript for a review! Certainly, authors should expand their review including the QSAR studies, for instance starting or by including the themes related the pharma and nutraceutical (Mini reviews in medicinal chemistry 12 (6), 467-476; Scientia pharmaceutica 78 (2), 233-248; Monatshefte für Chemie-Chemical Monthly 141 (5), 589-597) and related topics. Then the paper can enter the peer-review.

Answer: We appreciate the reviewer’s comment. In accordance with the editor’s comment, we added references #65 and #66 (page 8 in the new manuscript).

We hope that these changes are sufficient to satisfy the editor and the reviewer and that this manuscript will now be acceptable for publication in Int J Mol Sci.

Thank you for your consideration of our paper.

Sincerely yours,

Reviewer 2 Report

The paper is considerable improved and is susceptible to publication, yet after including additional references relating topological and QSAR studies on PAHs, since enlarging the appearance of a Review of the present otherwise worthy contribution: Current Organic Chemistry 17 (23), 2816-2830; Current Organic Chemistry 17 (23), 2845-2871. Once these references and related description are included, the paper can be re-checked for similarity and recommended for publication.

Round 2

Reviewer 1 Report

The article can be accepted in present form.